# Excessive Na-Doped La$_{0.75}$Sr$_{0.25}$Cr$_{0.5}$Fe$_{0.4}$Cu$_{0.1}$O$_{3-\delta}$ Perovskite as an Additional Internal Reforming Catalyst for Direct Carbon Dioxide-Ethanol Solid Oxide Fuel Cells

Mingfei Li [1], Jiangbo Dong [1], Zhengpeng Chen [1], Kairu Huang [2], Kai Xiong [3], Ruoyu Li [4], Mumin Rao [1], Chuangting Chen [1], Yihan Ling [4],* and Bin Lin [5],*

1   Guangdong Energy Group Science and Technology Research Institute Co., Ltd., Guangzhou 510000, China
2   Guangdong Huizhou Lng Power Co., Ltd., Huizhou 516000, China
3   Guangdong Energy Group Co., Ltd., Guangzhou 510000, China
4   School of Materials Science and Physics, China University of Mining and Technology, Xuzhou 221116, China
5   School of Mechanical and Electrical Engineering, University of Electronic Science and Technology of China, Chengdu 611731, China
*   Correspondence: lyhyy@cumt.edu.cn (Y.L.); bin@uestc.edu.cn (B.L.)

**Abstract:** Direct ethanol solid oxide fuel cells (SOFCs) are the most energy-efficient and low-carbon technology for renewable power generation from biomass fuels, but they are hindered by carbon deposition on the Ni-based cermet anode. In this work, excessive Na$^+$ dopant into La$_{0.75}$Sr$_{0.25}$Cr$_{0.5}$Fe$_{0.4}$Cu$_{0.1}$O$_{3-\delta}$ (LSCFC) perovskite was used as an additional internal reforming catalyst for direct carbon dioxide-ethanol SOFCs. Excessive Na$^+$-doped LSCFC (N-LSCFC) demonstrated great potential in promoting electrochemical performance and internal reforming process fueled by carbon dioxide-ethanol mixture, because more oxygen vacancies and the precipitated Cu nano catalyst were helpful for the improvement of internal reforming and carbon tolerance. Electrochemical investigations proved that the vertical-microchannel anode supported the single cells using the N-LSCFC-Gd$_{0.1}$Ce$_{0.9}$O$_{2-\delta}$ (GDC) internal reforming catalyst, showing a peak power density of 1044.41 and 855.56 mW/cm$^2$ at 800 °C fueled by H$_2$ and 50% CO$_2$-50% C$_2$H$_5$OH, respectively. The preceding results indicate that excessive Na$^+$ doping strategy into LSCFC as the additional internal reforming catalyst can improve the electrochemical performance and internal reforming process of direct carbon dioxide-ethanol SOFCs.

**Keywords:** solid oxide fuel cells; Ni-based anode; carbon resistance; internal reforming; carbon dioxide-ethanol

## 1. Introduction

Electricity can be directly produced from the chemical energy of fuels using solid oxide fuel cell (SOFC) technology, which has many advantages, such as high energy conversion efficiency and flexible fuel selectivity, leading to increasing attention [1,2]. Ethanol can be formed from renewable biomass fuels and there is no difficulty in its storage and transportation. The use of ethanol can greatly reduce the costs of SOFC commercialization. Ni-based cermet is an outstanding anode material for SOFCs due to its excellent comprehensive performance for fuel catalysis, thermochemical compatibility with electrolyte materials, and high conductivity [3]. However, carbon deposition will be generated on the active site of the Ni cermet anode after the long-term exposure of hydrocarbon fuels, resulting in the degradation of cell performance [4].

In recent years, many strategies have been proposed to reduce and eliminate carbon deposition on the active site of Ni cermet anode. One method is to add partial oxide gases into the fuels, such as water, carbon oxide, and oxygen into syngas for fuel reforming [5–7]. Accordingly, the dilution of fuel will cause a reduction in electrochemical performance. Some other researchers have explored various alternative methods to improve the carbon

resistance of Ni-based ceramics anodes, such as alloying Ni [8,9], Ni catalyst impregnated with metal oxide, and renormalized catalyst layers [10–12]. Among the internal reforming catalysts, $La_{0.6}Sr_{0.2}Cr_{0.85}Ni_{0.15}O_3$ is a robust catalyst for hydrocarbons due to its outstanding ability for dry reforming [12]. Recently, Li/K-doped perovskite materials have been applied to direct carbon SOFCs, showing excellent catalytic properties for gasification of solid carbon fuel [13–15]. Heping Xie et al. [16] reported partially replacing La sites with Li in $La_{0.7}Sr_{0.3}Fe_{0.8}Ni_{0.2}O_{3-\delta}$ perovskite oxide and found that Li doping could increase oxygen vacancies in the lattice and improve oxygen diffusion properties. The Fe-Ni alloy and $LaCO_3$ formed on the surface effectively improved the catalytic activity and stability of the electrode. Shao et al. [17] reported that the modification of anode with alkaline elements can effectively reduce the generation of anode carbon deposition in a short time. A novel perovskite material ($Li_{0.33}La_{0.56}TiO_3$) with alkaline elements (Li) was proposed, which can provide a stable Li source to compensate the loss of surface Li [17]. Several studies have clarified the reaction mechanism of ethanol dry reforming, generated carbon deposition, and optimized fuel gas composition and catalyst selection. They have also shown the impact of the catalyst, thermodynamics, and operating conditions on the process [18–20]. The yield of dry reforming equilibrium products has been shown to depend on the reaction temperature in the thermodynamic equilibrium analysis of the ethanol dry reforming process. The findings show that the content of solid carbon is dramatically reduced at a certain temperature [21,22].

It seems that partially doping with alkaline-earth metal at A sites is a good candidate for anode reforming to solve the problem of carbon deposition. In this work, excessive Na-doped $La_{0.75}Sr_{0.25}Cr_{0.5}Fe_{0.4}Cu_{0.1}O_{3-\delta}$ (N-LSCFC) perovskite as an internal reforming catalyst is applied to promote the coke tolerance of direct carbon oxide-ethanol SOFCs with a suitable volume ratio (1:1), where excessive Na provides rich acidic sites and the precipitation of Cu nanometal after hydrogen reduction could be beneficial to maintain the ability of carbon tolerance.

## 2. Result and Discussion

As shown in Figure 1a, the N-LSCFC oxide exhibits a single-phase cubic perovskite structure, which is in agreement with some previous crystal structure analytical results [23,24]. Figure 1b exhibits an enlarged section of the XRD patterns (32–33°). As Na is incorporated, these reflection angles gradually increase, suggesting lattice contraction occurs upon the introduction of Na ions. The XPS analysis of the binding energies of Na1s and O1s of the LSCFC and N-LSCFC is provided in Figure 2. The XPS spectra of Na1s of the N-LSCFC catalyst after being sintered at 1250 °C for 5 h is exhibited in Figure 2a. The presence of Na1s is obviously observed in the N-LSCFC sample compared to the LSCFC sample, indicating that Na completely enters the lattice of the LSCFC. Figure 2b,c display the O1s spectra of the LSCFC and N-LSCFC samples following calcination at 1250 °C for 5 h. The O1s is composed of lattice oxygen (530.1 eV), oxygen vacancy (531.4 eV), and absorbed oxygen (532.6 eV) [25]. The N-LSCFC sample has a large amount of oxygen vacancies, which are derived from the doping of Na in the LSCFC.

The N-LSCFC and GDC samples were mixed at a 1:1 weight ratio and sintered at 1250 °C for 5 h to evaluate the high-temperature chemical compatibility of N-LSCFC and GDC. As shown in Figure 3a, only the cubic structure of the N-LSCFC and the fluorite structure of the GDC are available, and no impurity phase is found, which indicates that N-LSCFC and GDC have good chemical compatibility below 1250 °C. The N-LSCFC was treated at 800 °C for 5 h under hydrogen and carbon dioxide atmosphere, respectively, to explore the atmospheric stability of the material. The XRD results after hydrogen reduction show that the N-LSCFC still maintains the original single perovskite structure (Figure 3b). The difference is that the precipitation of Cu alloy nanoparticles at B-site can be clearly observed, thus improving the catalytic performance. Similar phenomena have been reported in the literature [26]. The results of the sample treated with carbon dioxide show that, although the main perovskite structure of the N-LSCFC is still maintained, the

formation of an additional diffraction peak can be detected. The additional diffraction peak belongs to SrO (PDF # 27-1304), and some reports have been reported [16].

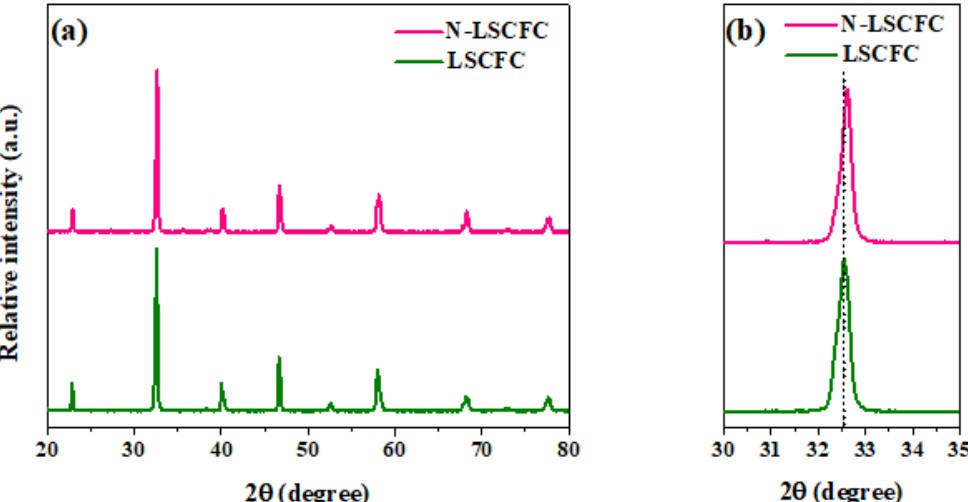

**Figure 1.** Room temperature XRD patterns. (**a**) LSCFC and N-LSCFC perovskites sintered at 1250 °C in air, and (**b**) magnification peak located at 32–33°.

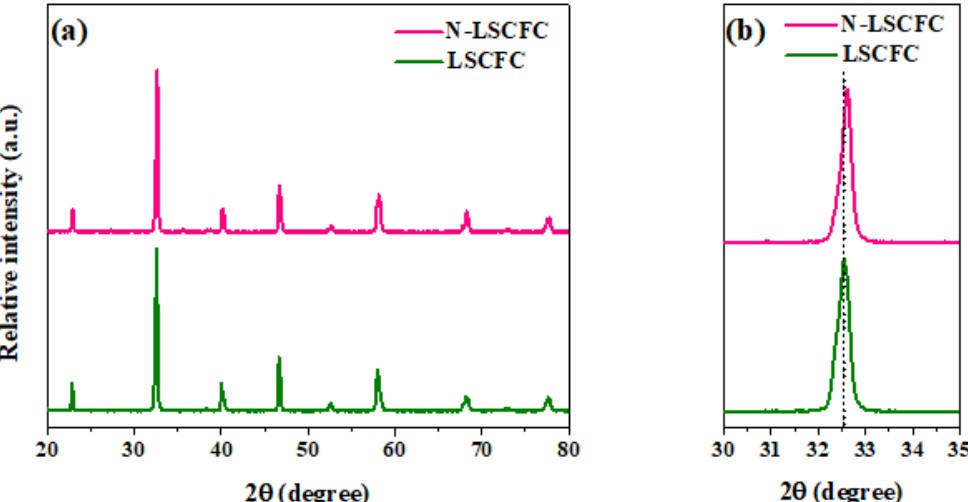

**Figure 2.** (**a**) XPS spectra of Na1s for LSCFC and N-LSCFC, and XPS spectra of O1s for (**b**) LSCFC and (**c**) N-LSCFC.

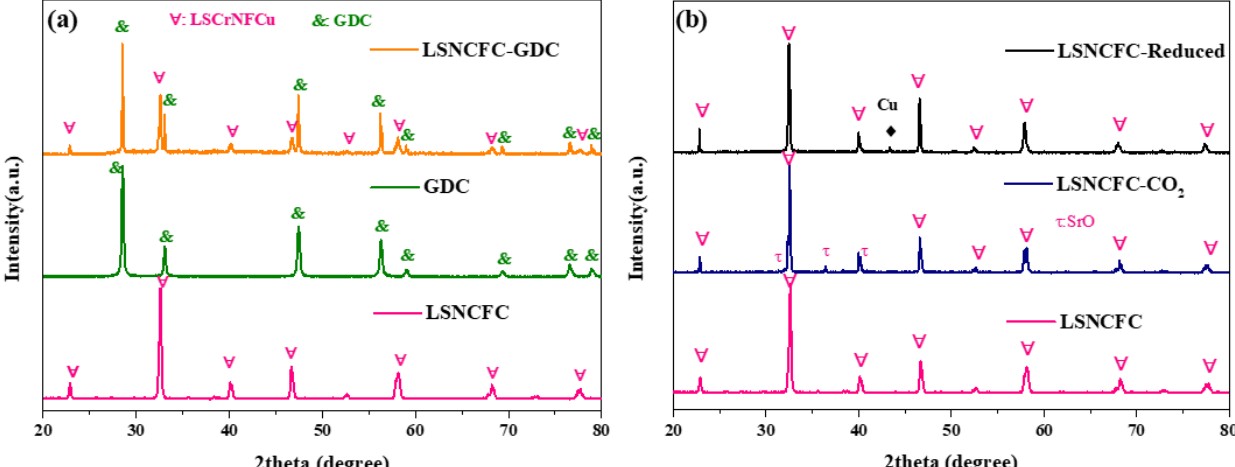

**Figure 3.** XRD patterns of (**a**) N-LSCFC, GDC, and N-LSCFC-GDC; (**b**) atmospheric treatment of $H_2$ and $CO_2$ for N-LSCFC.

The thermodynamics of 50% $CO_2$-50% $C_2H_5OH$ reforming reaction was calculated using the HSC software; the results are shown in Figure 4a. According to previous reports, ethanol dehydrogenation results in the formation of acetaldehyde on the metal surfaces and alkali sites, while ethylene generated by the reaction occurs at the acid site and is subsequently polymerized into carbon. [27,28]. The intermediate acetaldehyde is further broken down into $CH_4$ and CO or undergoes dry reforming with $CO_2$ to produce CO and $H_2$. Meanwhile, during the reforming process, ethanol can generate a certain amount of CO, $H_2$, $CO_2$, and $CH_4$. The breakdown of $CH_4$ may be the cause of carbon formation in this process. On the one hand, water–gas shift reaction may occur between CO and $H_2O$ at low reaction temperature, resulting in a decrease in CO selectivity and an increase in $H_2$ production. On the other hand, high reaction temperature will improve CO selectivity, owing to the development of a reverse water–gas shift reaction [29]. In this study, various gas compositions in the tail gas of the single cell with N-LSCFC as reforming catalyst in 50% $CO_2$-50% $C_2H_5OH$ atmosphere were detected; the results are shown in Figure 4b.

After cell test, the SEM cross-section microstructure was analyzed; the results are depicted in Figure 5a. It can be seen that the interfaces of all cell components are in fine contact. The NiO-YSZ anode has a considerable channel since the dendritic anode support is made by stainless steel auxiliary phase inversion, which can foster favorable conditions for gas mass transfer. The characteristics of each component of the single cell can be found, and the YSZ electrolyte is relatively dense after sintering (16 μm). This is also proven by the open-circuit voltage (OCV) of the single cell in $H_2$. A GDC barrier with a thickness of about 5 μm was added to prevent the reaction between PBSCF and YSZ at high temperature. The N-LSCFC-GDC renormalized catalyst was filled in the dendritic channel as a promising candidate for carbon tolerance, as shown in Figure 5b. The depth of the N-LSCFC-GDC in the dendrite channel is about 100 μm, which achieves the expected experimental assumption. EDS demonstrates the distribution of the renormalized catalyst elements and the presence of Na (Figure 5c,d).

Figure 6 illustrates the electrochemical performance of the single cell with N-LSCFC-GDC internal reforming catalyst in $H_2$ and 50% $CO_2$-50% $C_2H_5OH$ atmosphere, respectively. The OCV comparison of the single cell with reforming layer at different temperature is displayed in Figure 6c. Compared to the theoretical OCV, the OCV of the cell under different fuels is within the acceptable range, indicating that there is no problem in the cell preparation and testing process. The maximum power density (MPD) of the single cell with $H_2$ as fuel is 1044.41, 760.9, 489.2, and 275.57 mW/$cm^2$ at 800–650 °C, respectively. While the MPD of the single cell with ethanol carbon dioxide as fuel is 855.56, 582, 359.06, and 200.64 mW/$cm^2$ at 800–650 °C, respectively. Clearly, the MPD of a single cell in the $H_2$ atmosphere is slightly higher than that in the 50% $CO_2$-50% $C_2H_5OH$ atmosphere at different temperatures. This is

mainly due to the $CO_2$ reforming $C_2H_5OH$, and its electrochemical oxidation is still higher than that of hydrogen. Interestingly, when 50 vol.% ethanol is used as fuel in a single cell, the cell shows no obvious concentration polarization due to insufficient fuel supply, indicating that the addition of the anode reforming catalyst has little effect on the fuel mass transfer of the dendritic anode. However, this polarization may be clearly seen in the single cell created using dry-pressure method at the same or higher fuel partial pressure [30].

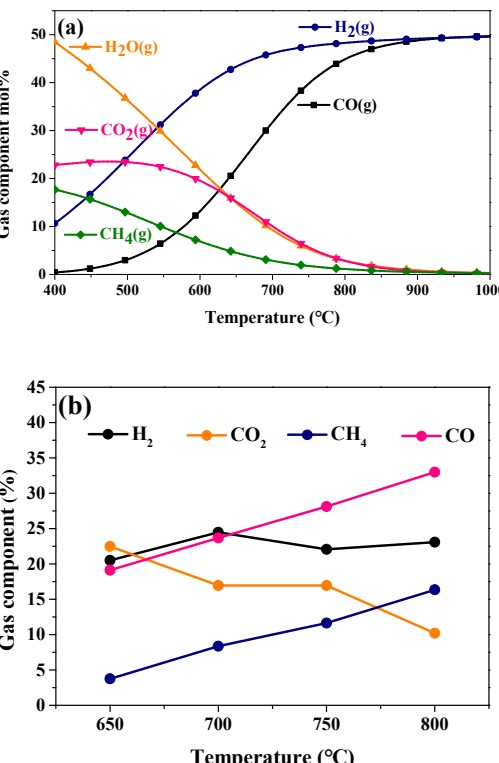

**Figure 4.** (**a**) the thermodynamic calculation of 50% $CO_2$-50% $C_2H_5OH$ dry reforming reactions; (**b**) gas composition of single cell with N-LSCFC reforming catalyst from 800 °C to 650 °C in 50% $CO_2$-50% $C_2H_5OH$ atmosphere.

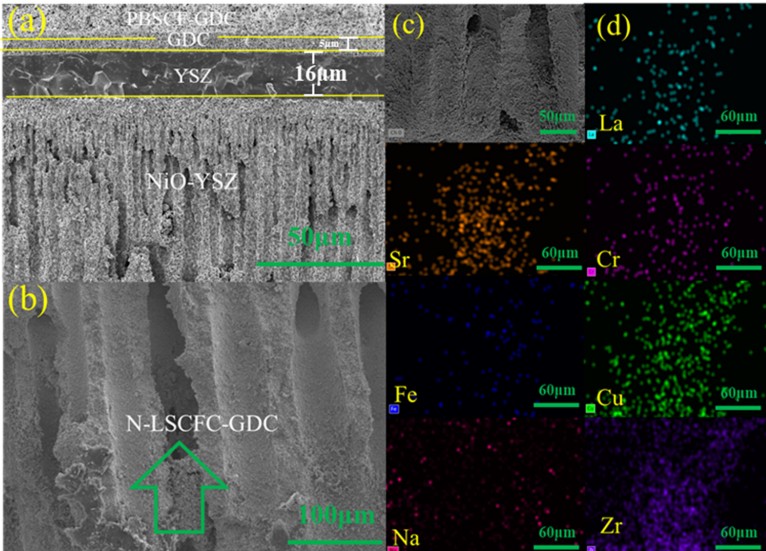

**Figure 5.** SEM microstructure and EDS element analysis of single cell with N-LSCFC-GDC reforming catalyst: (**a**) triple structure; (**b**) dendritic anode with N-LSCFC-GDC reforming catalyst; (**c**) image for EDS; and (**d**) elemental mappings for (**c**).

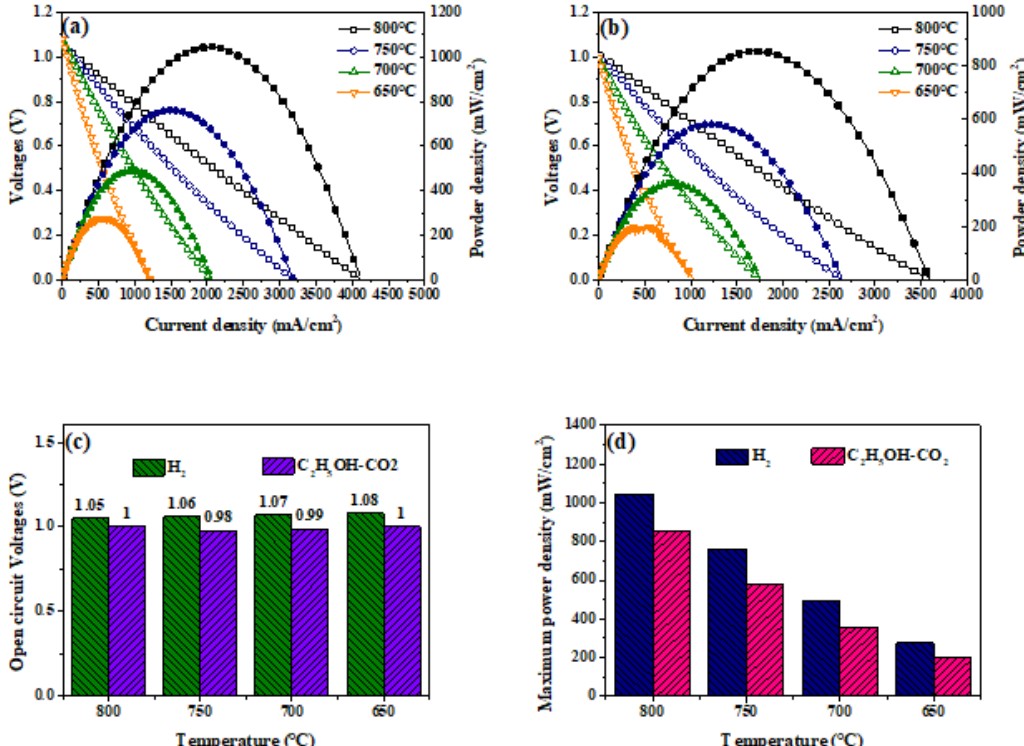

**Figure 6.** I-V-P curves of single cell with N-LSCFC-GDC reforming catalyst from 800 °C to 650 °C in (**a**) $H_2$(~3% $H_2O$) and (**b**) $C_2H_5OH$-$CO_2$ atmosphere; (**c**) OCVs and (**d**) MPD of single cell with N-LSCFC reforming catalyst.

The electrochemical impedance spectral (EIS) curves of the dendritic cell with N-LSCFC-GDC internal reforming catalyst in $H_2$ and 50% $CO_2$-50% $C_2H_5OH$ atmosphere are shown in Figure 7a,b. The first intersection point on the X axis represents the ohmic impedance ($R_o$), and the second intersection point represents the total impedance ($R_t$). The difference between these two values is the polarization impedance ($R_p$). When 50% $CO_2$-50% $C_2H_5OH$ is used as the fuel, the $R_o$ of the single cell is higher than that of the $H_2$. The same experimental results can be found in previous literature [30]. As the temperature decreases, the Rp of the single cell gradually increases, and the increase in 50% $CO_2$-50% $C_2H_5OH$ atmosphere significantly increases, as shown in Figure 7c. This may be due to the difficulty of electrochemical oxidation of the 50% $CO_2$-50% $C_2H_5OH$ fuel at low temperatures. The activation energies in $H_2$ and in 50% $CO_2$-50% $C_2H_5OH$ are about 93.69 kJ/mol and 120.05 kJ/mol, respectively, indicating that the anode $R_p$ of a single cell with 50% $CO_2$-50% $C_2H_5OH$ as fuel is greater than that of a cell fueled by $H_2$.

The stability of the single cell with N-LSCFC-GDC internal reforming catalyst at 750 mV in 50% $CO_2$-50% $C_2H_5OH$ atmosphere was measured; the results are shown in Figure 8a, with no obvious performance degradation. The carbon deposition on the anode surface of the single cell was scanned after the anode had been exposed to 50% $CO_2$-50% $C_2H_5OH$ atmosphere for 15 h, as shown in Figure 8b. Two coke-related Raman characteristic peaks, 1592 $cm^{-1}$ and 1348 $cm^{-1}$, are detected [31]. The G-peak reflects in-plane vibrational carbon atoms, while the D-band originates from the disorder of the pyrochlore structure. A large peak intensity ratio I(D)/I(G) indicates a low degree of carbon graphitization [32], which indicates that LSNCFC is a promising fuel reforming material for SOFCs.

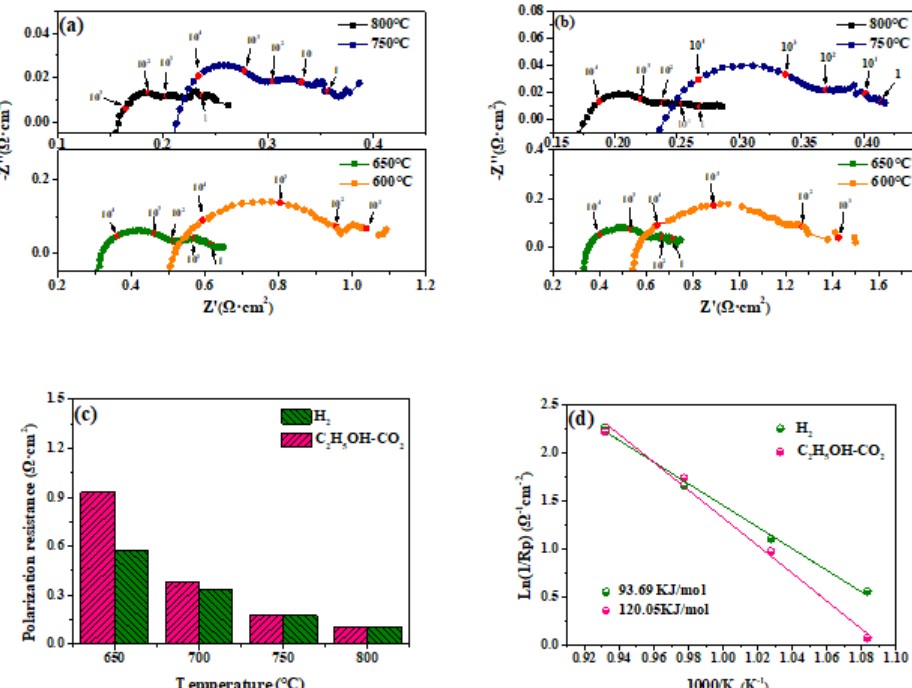

**Figure 7.** EIS of single cell with N-LSCFC-GDC reforming catalyst from 800 °C to 650 °C in (**a**) $H_2$ (~3% $H_2O$) and (**b**) $C_2H_5OH$-$CO_2$ atmosphere; (**c**) $R_p$ comes from impedance spectra; (**d**) Arrhenius's plots of the simulated $R_p$ values from (**c**).

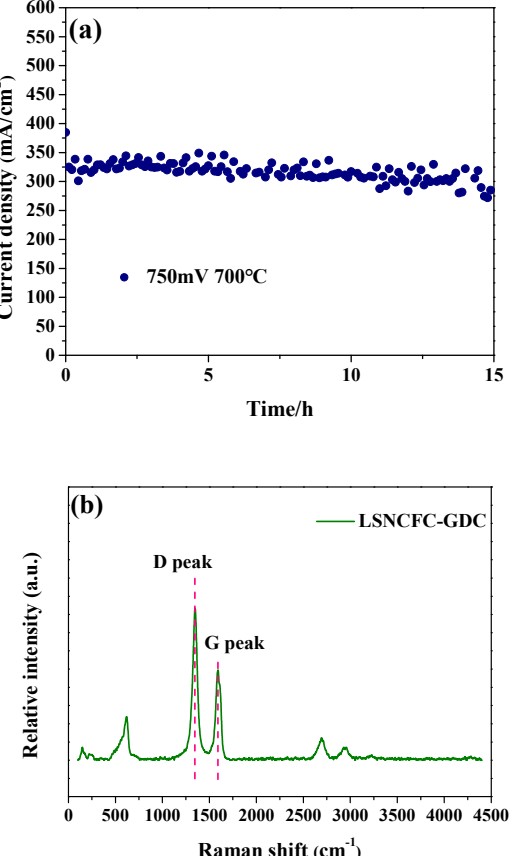

**Figure 8.** (**a**) The stability of single cell exposed to 50% $CO_2$-50% $C_2H_5OH$; (**b**) Raman spectra curves of single cell with N-LSCFC-GDC reforming catalyst after exposure to 50% $CO_2$-50% $C_2H_5OH$ at 700 °C for 15 h.

### 3. Experiment

An excessive Na-doped $La_{0.75}Sr_{0.25}Cr_{0.5}Fe_{0.4}Cu_{0.1}O_{3-\delta}$ (N-LSCFC) anode reforming catalyst was synthesized using citrate acid-ethylene diamine tetraacetic acid (EDTA) through a self-propagating combustion method. In stoichiometric amounts, $La(NO_3)_3$, $Sr(NO_3)_2$, $NaNO_3$, $Cr(NO_3)_3$, $Fe(NO_3)_3 \cdot 9H_2O$, and $Cu(NO_3)_2 \cdot 3H_2O$ were employed as the initial starting materials in the fabrication process. Citrate acid and EDTA played the role of complexion and combustion, and their molar ratios with metal ions were 1:1:0.8. Firstly, after the addition of the metal ion precursors to deionized water according to the weight of the stoichiometry, the solution was stirred at 80 °C for 2 h to completely dissolve the metal ion starting material. Next, citrate acid and EDTA were poured into a beaker to form a uniform aqueous solution. Ammonia could be used to adjust the homogeneous solution's PH to 7~8. The above solution was continuously heated by a resistance furnace, with the evaporation of deionized water to form a viscous gel accompanied by combustion. Dark brown burning ashes placed in the muffle furnace were further sintered at 1250 °C for 5 h to remove residual impurities. A similar preparation process had been presented in our previous work [33].

In this work, the dendritic anode was prepared using phase-inversion molding technology [34]. The anode slurry was composed of 68.2 wt.% of NiO-YSZ powder, 0.48 wt.% polyvinylpyrrolidone (PVP), 31.32 wt.% of polyether sulfone (PES), and 1-Methyl-2-pyrrolidinone (NMP) mixed solution. The weight ratio of NiO to YSZ powder was 60:40. PESF (17.7 wt.% of NMP) was dissolved in NMP to form a uniform solution. After ball milling, the slurry was vacuumed for 30 min at room temperature for the preparation of the anode substrate. The slurry was injected into the upper and lower layers of stainless mold, and the middle was separated by a stainless mesh with a hole diameter of 150 microns.

Tape water was used as coagulant. The achievement of raw cermet was sintered at 1050 °C for 2 h to acquire the anode substrate. Subsequently, the YSZ slurry was spin coated on the specific side of the anode substrate and then sintered at 1400 °C.

GDC was fabricated by screen printing and sintered at 1250 °C. The fabrication process of the YSZ and GDC slurries were introduced in detail in our previous work [35]. The cathode slurry of $PrBa_{0.5}Sr_{0.5}Co_{1.5}Fe_{0.5}O_{5+\delta}$ (PBSCF)-GDC (6:4) was deposited on the GDC and then sintered at 950 °C for 3 h. The cathode's active area was 0.24 $cm^2$. The mixture of N-LSCFC, GDC (6:4), and 5% terpineol ethyl cellulose was grounded to obtain the anode catalyst slurry. This slurry was brushed on the dendritic holes and sintered at 1250 °C for 3 h to obtain an excellent combination with straight hole structure. The sintering of the anode catalyst was prior to that of the cathode.

The characteristic of powders, the chemical compatibility, and the atmosphere were analyzed using an X-ray diffractometer (XRD, Bruker 8ADVANCE, Bremen, Germany). X-ray photoelectron spectroscopy (XPS, ESCALAB QXi, Thermo Fisher Scientific, Waltham, MA, America) analysis was used to detect the oxidation states of elements in the LSCFC and N-LSCFC. The electrochemical dates were performed using an electrochemical workstation (Squidstat Plus, Admiral Instruments, Tempe, AZ, America). A scanning electron microscope (SEM, Gemini-300, ZEISS, Aalen, Germany) was used to investigate the single cell's microstructure. An X-ray spectroscopy (Oxford Instruments AZtecEnergy, Abingdon, England) was employed to research the single cell's energy dispersive spectroscopy (EDS) mapping. Carbon decomposition on the anode was explored using a Raman spectroscopy (SENTERRA, Bruker, Bremen, Germany). The tail gas was tested using a gas chromatography (Shimadzu GC-2014, Kyoto, Japan).

### 4. Conclusions

Excessive $Na^+$ dopant into $La_{0.75}Sr_{0.25}Cr_{0.5}Fe_{0.4}Cu_{0.1}O_{3-\delta}$ perovskite is used as an additional internal reforming catalyst for direct carbon dioxide-ethanol SOFCs, because more oxygen vacancies and the precipitated Cu nano catalyst are helpful for the improvement of internal reforming and carbon tolerance. The internal reforming catalyst N-LSCFC-GDC is coated on the dendritic anode surface to improve the carbon tolerance of Ni-based SOFCs

in direct 50% $CO_2$-50% $C_2H_5OH$ atmosphere. The single cell with the internal reforming catalyst N-LSCFC-GDC owns excellent electrochemical performance (855.56 mW/cm$^2$, 800 °C) in 50% $CO_2$-50% $C_2H_5OH$ due to the dendritic anode structure and the existence of the N-LSCFC-GDC reforming catalyst; Excessive Na$^+$ doping strategy into LSCFC as the additional internal reforming catalyst can improve the electrochemical performance and internal reforming process of direct carbon dioxide-ethanol SOFCs.

**Author Contributions:** Conceptualization, M.L. and M.R.; methodology, K.H.; software, R.L.; validation, M.L, J.D. and Z.C.; formal analysis, C.C.; investigation, C.C.; resources, Z.C.; data curation, K.X.; writing—original draft preparation, M.L. and K.H.; writing—review and editing, Y.L. and B.L.; visualization, K.X.; supervision, Z.C.; project administration, M.R. and Y.L.; funding acquisition, Y.L. All authors have read and agreed to the published version of the manuscript.

**Funding:** This research was supported by the Ministry of Science and Technology of China (No. 2021YFE0100200), the Pakistan Science Foundation (PSF) Project (No. PSF/CRP/18thProtocol (01)), the National Natural Science Foundation of China (No. 52272257), and the Fundamental Research Funds for the Guangdong Provincial Key Research and Development Program-China (2022B0111130004).

**Data Availability Statement:** The data presented in this study are available on request from the corresponding author.

**Conflicts of Interest:** The authors declare no conflict of interest.

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
