# Peer review of "Excessive Na-Doped La0.75Sr0.25Cr0.5Fe0.4Cu0.1O3-δ Perovskite as an Additional Internal Reforming Catalyst for Direct Carbon Dioxide-Ethanol Solid Oxide Fuel Cells"

_catalysts, doi:10.3390/catal12121600_

Round 1

Reviewer 1 Report

Li et al., studied the solid oxide fuel cells (SOFCs) and analyzed the effect of Na+ dopant on

 La0.75Sr0.25Cr0.5Fe0.4Cu0.1O3-δ 18 (LSCFC) perovskite. The manuscript is suggested to be accepted after the following issues are addressed;

1)       In XRD, the authors should index all the XRD peaks and Rietveld refinement.

2)      The authors should add more information about the XRD such as particle size etc.

3)      Rescaled the EDX image, it is hard to read the scale and percentage of elements

4)      What’s about the thickness, did the author confirm with some equipment

5)       What’s about the leakage current

6)      Many spelling and formatting typos in this paper, and we hope the authors should check and revise them thoroughly.

Reviewer 2 Report

Journal: Catalysts (ISSN 2073-4344)

Manuscript ID: catalysts-2073589

Type: Article

Title: Excessive Na-doped La0.75Sr0.25Cr0.5Fe0.4Cu0.1O3-δ perovskite as an additional internal reforming catalyst for direct carbon dioxide-ethanol solid oxide fuel cells.

Authors: Mingfei Li , Jiangbo Dong , Zhengpeng Chen , Kairu Huang , Kai Xiong , Ruoyu Li , Mumin Rao , Chuangting Chen , Yihan Ling * , Bin Lin *.

[1]         Keywords: 5 words

[2]         Introduction: write the perspective of the present work carefully.

[3]         Results and Discussion:

Why the author didn’t measure the other mechanical properties such as compression strength of the samples?

Why the author didn’t measure the other optical properties of the samples?

[4]   What does it add to the subject area compared with other published material?

[5]   Do you consider the topic original or relevant in the field? Does it address a specific gap in the field?

[6]   What is the main question addressed by the research?

[7]         References: cite the following recent references

DOI: https://doi.org/10.1088/1742-6596/1795/1/012050

DOI: https://hal.archives-ouvertes.fr/hal-02443179

Best Regards

Round 2

Reviewer 1 Report

Accept in present form